# Alcohol Consumption Patterns: A Systematic Review of Demographic and Sociocultural Influencing Factors

**DOI:** 10.3390/ijerph19138103

**Published:** 2022-07-01

**Authors:** Abd Alghani Khamis, Siti Zuliana Salleh, Mohd Sayuti Ab Karim, Noor Ashikin Mohd Rom, Shamini Janasekaran, Aida Idris, Rusdi Bin Abd Rashid

**Affiliations:** 1Department of Mechanical Engineering, Faculty of Engineering, Universiti Malaya, Kuala Lumpur 50603, Malaysia; a-a-f-k@hotmail.com (A.A.K.); zuliana_2014@hotmail.com (S.Z.S.); 2Centre of Advanced Manufacturing and Material Processing (AMMP), Department of Mechanical Engineering, Faculty of Engineering, Universiti Malaya, Kuala Lumpur 50603, Malaysia; 3Faculty of Management, Multimedia University, Cyberjaya 63100, Malaysia; ashikin.rom@mmu.edu.my; 4Centre for Advanced Materials and Intelligent Manufacturing, Faculty of Engineering, Built Environment & IT, SEGi University Sdn Bhd, Petaling Jaya 47810, Malaysia; shaminijanasekaran@segi.edu.my; 5Department of Management, Faculty of Business and Economics, Universiti Malaya, Kuala Lumpur 50603, Malaysia; aida_idris@um.edu.my; 6Department of Psychological Medicine, Faculty of Medicine, Universiti Malaya, Kuala Lumpur 50603, Malaysia; rusdi@um.edu.my

**Keywords:** alcohol consumption patterns, physical and mental health, demographics, social factors

## Abstract

Background: Excessive alcohol consumption has negative effects not only on the drinkers’ health but also on others around them. Previous studies suggest that excessive alcohol consumption can be related to a combination of factors such as age, family background, religiosity, etc. Investigating and clarifying these roots of alcohol consumption is crucial so that the right type of interventions can be designed in a specific and targeted manner. Objectives: This work was conducted as a systematic review to reveal the factors associated with alcohol consumption and to heighten the understanding of the differences among various communities and segments of the population regarding their usage of alcohol. Data sources: A systematic search of Web of Science, PubMed, ScienceDirect, and Google Scholar was performed. Results: Forty-five studies were included in the review after excluding irrelevant records and duplicates. Conclusions: Alcohol consumption patterns can be associated with several factors related to communities and individuals, and our review revealed demographic factors, including age and proximity to alcohol outlets, as well as social factors, including family background, socioeconomic background, and religious influence. These findings can be used to establish a guideline for further studies in understanding alcohol consumption patterns among individuals according to their personal characteristics and sociocultural backgrounds.

## 1. Introduction

Alcohol existence and consumption can be traced as far back as 7000 BC [1]. A number of medical studies have reported the positive effects of moderate alcohol consumption. For instance, it can prevent certain diseases and medical conditions, such as a heart attack [2]. It has also been observed that individuals can use alcohol as an aiding tool to control social conditions. For instance, Hajek et al. [3] found an association between decreased loneliness, higher life satisfaction, and less perceived stress with those who reported occasional and daily drinking. However, alcohol also possesses both toxic and intoxicating properties. Alcohol is a toxic substance that is foreign to the body (not produced by the body), and it can lead to serious poisonous effects, especially when taken in high concentrations [4]. In general, the harmfulness of alcohol consumption can be related to the total volume of irregular heavy drinking [5,6]. Furthermore, people who drink frequently in licensed establishments are more likely to be harmed by other’s drinking [7].

Awareness about the negative effects of alcohol consumption on physical and mental health has increased in recent decades. The main reason for this is that alcohol not only harms individual drinkers but also the wellbeing of their families and communities. In general, excessive alcohol consumption accounts for 5.1% of global diseases and injuries [8]. According to an investigation based on gender, harmful drinking accounts for 7.1% and 2.2% of global diseases for men and women, respectively [9]. Furthermore, alcohol consumption is also responsible for 10% of all deaths among people aged 15–49 [10]. Besides that, alcohol is the leading cause of premature death and disability of newborn babies [11]. Alcohol consumption also has other negative effects on individual performance, such as slower responses, particularly as a result of an alcohol hangover [12]. This was also observed by Aas et al. [13], who reported that employees’ consumption of alcohol was related to their performance both at the workplace and outside.

Furthermore, alcohol consumption can cause mortality in offspring due to suicide or violence. Landberg et al. [14] reported that the threat of violent death has been increasing among boys whose fathers are frequent consumers, and the threat of suicide increases in the highest category of fathers’ consumption. Heavy alcohol consumption and extreme weekly binge drinking have a larger effect on cognitive decline in an adult’s life [15]. Alcohol consumption adversely affects consumers’ driving performance, as it also has a degradation effect on vision [16]. Additionally, socioeconomic inequalities could result in inequal alcohol-related harm, despite similar consumed quantities [17,18]. Although a number of interventionist approaches have been implemented by governments to lower the rate of alcohol consumption, life-threatening situations due to alcohol abuse still prevail [19].

To address the problems associated with alcohol abuse, it is necessary to investigate and understand the reasons for alcohol consumption among different communities and segments of the population. Survey investigations regarding alcohol consumption have been widely reported [20,21,22]. According to the literature, alcohol consumption patterns can be related to life events [23], as well as demographic and sociocultural factors [24]. For instance, it can be related to age [25,26,27,28], socioeconomic background [29], and family background [30]. 

This review paper examines the different approaches to study alcohol consumption among different communities and segments of the population. It also discusses the relationship between risky alcohol consumption and various personal demographics and sociocultural factors among individuals and communities. A qualitative hypothesis was formed to drive the search of this review and was broken down into several questions as follows:

Q1: How can different factors among individuals and communities such as age, family background, or socioeconomic conditions be used as predictors of alcohol consumption patterns?

Q2: Does the extant literature include an adequate investigation of the abovementioned associations?

Q3: What are the strengths and weaknesses of earlier studies that can serve as a guide for future related research?

## 2. Methodology

Our research was conducted mainly on the Web of Science, PubMed, ScienceDirect, and Google Scholar search engines. Other sources of data were also used, including the World Health Organization’s formal website and manual tracking of cited records. The study focused on acquiring published articles related to alcohol use and related influencing factors in general. The query used to obtain the records used in this review was a combination of alcohol-related keywords and influencing factor keywords, as shown in Table 1. These combinations were (“Alcohol misuse” OR “Alcohol consumption” OR “Alcohol consumption patterns”) AND (“Socioeconomics” OR “Age” OR “Family” OR “Influencing factors” OR “Proximity to alcohol outlets” OR “Alcohol outlets” OR “Religiosity influence” OR “Religion”).

The eligibility for study inclusion was limited to studies that were published in the English language, scored 50% and above using the Quality Assessment Tools developed by the National Heart, Lung, and Blood Institute, and published within the past six years (2017–2022). We used EndNote X8 during the entire screening process and Mendeley as a citing manager. Screening the records focused on collecting information about the causes of more alcohol consumption. We screened the records for the inclusion criteria where only records concerning the first research question specified earlier were chosen. The other two research questions were answered later based on the results of the quality assessment.

According to the PRISMA guidelines, a systematic review is a survey that utilizes express, orderly techniques to group and integrate discoveries of studies that address an obviously figured-out question [31]. We followed the PRISMA throughout the screening process and the designing of the methodology. Figure 1 shows the abstract of the PRISMA flow of the review. We began by designing the research, then collecting all identified records and extracting the data. At a later stage, an assessment of the quality of the studies was performed, whereby the National Heart, Lung, and Blood Institute’s Study Quality Assessment Tool was used to assess the quality of the studies involved in the qualitative synthesis [32]. A. Khamis and S.Z. Salleh were involved in data extraction and collection for all identified records, and A. Khamis worked on the assessment of the study quality independently while S.Z. Salleh reviewed the assessment. No meta-analysis was performed due to the variety of measures and outcomes.

**Table 1 ijerph-19-08103-t001:** Quality assessment of the observational studies.

Category	N	Association	Study	1	2	3	4	5	6	7	8	9	10	11	12	13	14	Total (%)
Physical Exposure (Access)	1	Positive	(Morrison et al., 2019) [33]	√	√		√		√	√	√	√		√			√	9 (64.29)
	2	Positive	(Seid et al., 2018) [34]	√	√	√	√			√	√	√		√	√		√	10 (71.42)
	3	Positive	(Lamb et al., 2017) [35]	√	√		√		√	√	√	√		√	√		√	10 (71.42)
	4	Positive	(Toornstra et al., 2020) [36]	√	√		√				√			√	√		√	7 (50)
	5	Positive	(Martins et al., 2020) [37]	√	√	√	√		√		√	√		√			√	9 (64.29)
	6	Positive	(Tschorn et al., 2021) [23]	√	√		√		√	√	√	√	√	√	√	√	√	12 (85.71)
	7	Positive	(Foster et al., 2017) [38]	√	√	√	√		√	√	√	√	√	√		√	√	12 (85.71)
	8	Positive	(Foster et al., 2020) [39]	√	√	√	√		√	√	√	√	√	√		√	√	12 (85.71)
	9	Positive	(Auchincloss et al., 2022) [40]	√	√		√		√	√	√	√	√	√			√	10 (71.42)
	10	Positive	(Cardoza et al., 2020) [41]	√	√		√	√		√	√	√		√			√	9 (64.29)
	11	Positive	(Colchero et al., 2022) [41]	√	√	√	√		√	√	√	√		√	√		√	11 (78.57)
	Score																	10.1 (72.14)
	1	neutral	(Mair et al., 2020) [42]	√	√		√		√	√	√	√		√			√	9 (64.29)
	Score																	9 (64.29)
Age	1	Early access predicts more drinking	(Kim et al., 2017) [43]	√	√	√	√		√	√	√	√	√	√		√	√	12 (85.71)
	2	Early access predicts more drinking	(Staff & Maggs, 2020) [44]	√	√	√	√	√	√	√	√	√	√	√		√	√	13 (92.86)
	3	Early access predicts more drinking	(Soundararajan et al., 2017) [45]	√	√		√			√	√	√		√			√	8 (57.14)
	4	Early access predicts more drinking	(Plenty et al., 2019) [46]	√	√	√	√		√	√	√	√	√	√			√	11 (78.57)
	Score																	11 (78.57)
	1	Aging reduces consumption quantity	(Listabarth et al., 2020) [47]	√	√		√				√	√		√			√	7 (50)
	2	Aging increase frequency but reduce quantity	(Chaiyasong et al., 2018) [48]	√	√	√	√	√		√	√	√		√			√	10 (71.42)
	3	Aging reduces consumption	(Cheah & Rasiah, 2017) [49]	√	√	√	√	√		√	√	√		√				9 (64.29)
	Score																	8.6 (61.42)
	1	Age of onset predict alcohol consumption	(Aguilar et al., 2022) [50]	√	√		√		√	√	√	√		√				8 (57.14)
	2	Age of onset predict alcohol consumption	(Islam, 2020) [51]	√	√		√			√	√	√		√	√		√	9 (64.29)
	3	Age of onset predict alcohol consumption	(Aiken et al., 2018) [52]	√	√		√		√	√	√	√	√	√	√	√	√	12 (85.71)
	Score																	9.6 (68.57)
Family background influence	1	Permissive Parenting authority—negative	(Mathialagan et al., 2017) [53]	√	√		√			√	√	√		√				7 (50)
	Score																	7 (50)
	1	Permissive Parenting authority—positive	(Dickens et al., 2018) [54]	√	√		√			√	√	√		√	√			8 (57.14)
	Score																	8 (57.14)
	1	Family instability—positive	(Boua et al., 2021) [55]	√	√		√			√	√	√		√			√	8 (57.14)
	2	Family instability—positive	(Sumskas & Zaborskis, 2017) [56]	√	√	√	√	√			√	√		√	√		√	10 (71.42)
	Score																	9 (64.29)
	1	Family exposure—positive	(Haeny et al., 2020) [57]	√	√		√		√	√	√	√		√	√		√	10 (71.42)
	2	Family exposure—positive	(Berent & Wojnar, 2021) [58]	√	√		√		√	√	√	√		√				8 (57.14)
	3	Family exposure—positive	(Tschorn et al., 2021) [23]	√	√		√		√	√	√	√	√	√	√	√	√	12 (85.71)
	4	Family exposure—positive	(Gitatui et al., 2019) [59]	√	√		√	√		√	√	√		√			√	9 (64.29)
	5	Family exposure—positive	(Talley et al., 2018) [60]	√	√		√		√	√	√	√		√	√	√		11 (78.57)
	Score																	10 (71.42)
	1	Family exposure—neutral	(Clare et al., 2019) [61]	√	√	√	√			√	√	√	√	√		√	√	11 (78.57)
	Score																	11 (78.57)
Socio-economic	1	Positive	(Najafi et al., 2020) [62]	√	√	√	√				√	√		√	√	√		9 (64.29)
	2	Positive	(Beard et al., 2019) [29]	√	√		√	√		√	√	√		√				8 (57.14)
	3	Positive	(Cardoza et al., 2020) [41]	√	√		√	√		√	√	√		√			√	9 (64.29)
	4	Positive	(Cheah & Rasiah, 2017) [49]	√	√	√	√	√		√	√	√		√				9 (64.29)
	5	Positive	(Mair et al., 2020) [42]	√	√		√		√	√	√	√		√			√	9 (64.29)
	6	Positive	(Chaiyasong et al., 2018) [48]	√	√	√	√	√	√	√	√	√		√			√	11 (78.57)
	Score																	9.2 (65.71)
	1	Negative	(Čihák, 2020) [63]	√	√		√		√	√	√	√	√	√	√			10 (71.42)
	2	Negative	(Khan & Shaw, 2021) [64]	√	√	√	√			√	√	√		√	√			9 (64.29)
	Score																	9.5 (67.85)
Religiosity	1	Negative	(Charro Baena et al., 2019) [65]	√	√		√			√	√	√		√	√			8 (57.71)
	2	Negative	(Bai, 2021) [66]	√	√	√	√			√	√	√		√	√	√	√	11 (78.57)
	3	Negative	(Yockey et al., 2020) [67]	√	√		√			√	√	√		√	√			8 (57.71)
	4	Negative	(McAuslan et al., 2020) [68]	√	√		√		√	√	√			√	√			8 (57.71)
	5	Negative	(Stauner et al., 2019) [69]	√	√		√			√	√	√		√				7 (50)
	6	Negative	(Rivera et al., 2018) [70]	√	√		√			√	√	√		√	√			8 (57.14)
	7	Negative	(Dickens et al., 2018) [54]	√	√		√			√	√	√		√	√			8 (57.14)
	8	Negative	(Agrawal et al., 2017) [71]	√	√	√	√		√	√	√	√		√	√	√		11 (78.57)
	9	Negative	(Kathol & Sgoutas-Emch, 2017) [72]	√	√		√		√	√	√	√		√				8 (57.14)
	10	Negative	(Thompson, 2017) [73]	√	√		√		√	√	√	√		√				8 (57.14)
	11	Negative	(Guo & Metcalfe, 2019) [74]	√	√	√	√		√	√	√	√	√	√	√	√	√	13 (92.86)
	Score																	8.9 (63.57)
Average total																		9.5 (67.85)

The systematically identified papers were assessed using The National Heart, Lung, and Blood Institute’s (NHLBI, July 2021) quality assessment tool to rate the quality of the evidence for the observational studies. The NHLBI tool assessed the quality of publication using several criteria, as listed below:
1 = Research questions clearly stated;2 = Study population clearly defined;3 = Participation rate ≥50%;4 = Subjects from the same population and inclusion/exclusion criteria specified;5 = Sample size justification included;6 = Exposure measured before outcome;7 = Sufficient timeframe to see an effect;8 = Different levels of exposure included;9 = Clearly defined exposure measures;10 = Exposures measured more than once;11 = Clearly defined outcome measures;12 = Outcome assessors blinded to exposure status;13 = Loss to follow-up ≤ 20%;14 = Confounders measured and adjusted for.

This review was conducted to explore the association between demographic and/or social characteristics and alcohol usage; consequently, publications that documented the effects of alcohol were excluded from the quality rating process. To avoid any possible bias in the quality ranking process, the studies were ranked based on the given criteria before sorting them into categories. The findings of each study were later sorted into categories to reveal the different associations. In case of finding conflicts, each kind of association was sorted individually within the category itself to be compared and thoroughly discussed later. If a study had multiple findings that can be sorted into different categories, the ranking of the study will be used in every relevant category, and the summary of the findings will be divided into different categories.

The ranking for criteria 1 and 2 was done straightforwardly just by reading through the paper and trying to identify the information regarding the research questions and the population, while, for criteria 3, the assessment was based on a deeper search. For instance, if the study was sampling data from a bigger study or survey, we looked for the source of these data and identified the original response rate of the initial sample and then ranked them accordingly. Additionally, the response rate for studies that were multi-nationals or sampling data from multiple sources was assessed by calculating the average response rate across all sources. A point for criteria 4 was given if the study specified the inclusion/exclusion criteria of the population or if the information about the sample was clear enough for the study to be done again. Studies that did not justify their sample size or provide a statistical power calculation were not given criteria 5. If the study included information about the exposure measurements before the outcome assessment and if the study was cross-sectional yet the exposure took time previously, such as when sampling from a religious affiliating university to examine the associations between religiosity and consumption, criteria 6 was given due to the fact that the exposure to religion was present before the study. Some other studies collected data from participants at a specific point of time without analyzing the exposure time frame; those studies did not get points for criteria 7. However, we looked into the studies more thoroughly, and in cases where the exposure took enough time to have an influence on the participant’s life before the study took place, then we gave that a point for criteria 7; for example, under religious influence, if the participants had enough time to be involved in the religion, were the proper age, or living with parents following that religion, then they got a point for criteria 7 under the assumption that the sample had enough time to be influenced by this exposure. Points for criteria 8, 9, 10, and 11 were assessed by looking for straight answers within the study texts. A point for criteria 12 was not given for studies that did not include information about the blinded assessors, as we assumed they were blinded for studies analyzing secondary data (e.g., government surveys); however, when both outcomes and exposures were assessed by the same survey and timeframe, a point for criteria 12 was not given. For criteria 13, the follow-up retention rate was retrieved by either looking through the original source of data for each study or from within the study itself if it was reported. Finally, criteria 14 was assessed by looking at the statistical analysis and the discussion of the results.

The influencing factors were extracted from the studies only if they were viewed as an independent variable and had statistical significance. The influencing factors were categorized into five categories: proximity to alcohol outlets (physical exposure), age exposure, family influence, and socioeconomic and religious influence. The proximity to alcohol outlets associated with alcohol consumption combined both on-premises and off-premises outlets and investigated the drinking patterns associated with the outlet density in general. The age exposure included the influence of early access to alcohol, the influence of aging on the consumption patterns, and the influence of the age of their first drink. Family influence included the effect of parenting authority, the effect of family instability, and familial exposure. In the socioeconomic exposure category, the direct influences of socioeconomic status at both the individual and collective levels were included. Finally, religious influences were just discussed in a general matter to reveal whether religiosity has an influence on the drinking profile without any comparisons between different religions.

## 3. Results

As indicated in the PRISMA diagram (Figure 1), we were able to collect 1296 records using the query mentioned above. The total number of reports and studies included in this review was 45 after removing the duplicates (using the EndNote automated tool), irrelevant records (by screening titles and abstracts), and records written in other languages but English (by screening the full texts). Most of the studies reported multiple findings; however, this did not affect our data extraction, since the study focused solely on the association between demographic/social factors and alcohol consumption patterns and hence extracted the related findings and included them in the appropriate category individually.

The overall percentage of the quality assessment criteria met for all the findings (n = 51) was 67.85%, with an average met criteria of 9.5 (Table 1). Criteria number 1 (research question clearly stated), number 2 (study population clearly defined), number 4 (clear population and inclusion/exclusion criteria), number 8 (different levels of exposure included), and number 11 (defined outcome measures) were satisfied by all the findings, while the most unsatisfied criteria were number 5 (sample size justification included), with only 10 studies satisfying it, criteria number 10 (exposures measured more than once), with 12 studies satisfying it, and criteria number 13 (follow-up with a loss less than 20%), with 13 studies satisfying it. These findings provide lessons for establishing better research designs in the future to avoid the risk of bias.

For the association with physical exposure to alcohol outlets, 12 studies were identified. Eleven studies reported a positive association with a higher density of alcohol outlets and an average quality of evidence score of 10.1 (72.14%), mostly lacking sample size justifications and follow ups, while the other study that did not report a positive association only reported neutral effects, concluding that an increased density of outlets did not show an increment in either heavy drinking or AUDIT scores. That study scored 9 points (64.29%) for the quality criteria.

For the age exposure category, 10 studies were identified. These studies were divided into three different types of associations: early access to alcohol, aging, and age of onset, with four studies identified for early access and three studies for aging and the age of onset. The highest overall score for the quality of evidence rank was for the early access association, satisfying 78.57% of the quality criteria on average, while aging scored the lowest average of 62.42%, and the age of onset scored 68.57% on average.

The family background influence was also backed up with 10 studies divided into three categories: parenting authority, family instability, and family exposure. For parenting authority, two studies were identified related to permissive parenting authority with conflicting findings; one was positively associated, with a 57.14% quality rank, compared to the other negatively associated with a 50% quality rank. Similarly, two studies were sorted into the family instability category, both studies had a positive correlation with alcohol consumption, with an average 64.29% quality rank. Finally, the remaining six studies were sorted into family exposure association; five were positively associated with predicting future alcohol consumption, with an average quality score of 71.42%, and one was neutral, with a quality of evidence score of 78.57%. A more thorough breakdown of these findings will be done in the Discussion section.

In the socioeconomic exposure category, the findings were either on a collective level (overall socioeconomic condition of a nation or a group) or at an individual level (personal monthly income or individual status of wealth). Both the collective level and individual level were used to show the associations; however, there was some conflict in the findings. We found six studies to be positively associated with alcohol consumption (65.71% average quality rank) and two negatively associated (67.85% average quality rank).

Finally, 11 studies were sorted into religious influence; all of them reported a negative association with alcohol consumption, with an average quality rank of 63.57%, most lacking criteria number 10 (exposure measured more than once) and all lacking criteria number 5 (sample size justification).

## 4. Discussion

As mentioned earlier, the roots of alcohol consumption in the community need to be investigated to reduce and treat excessive or irregular heavy drinking. Based on a systematic literature review, we can extend our current understanding of the factors associated with alcohol consumption. These factors can be categorized as either demographic (proximity and age) or social (family influence, socioeconomic background, and religious influence), as illustrated in Figure 2. The studies in the literature had a wide variety of outcomes, including frequency, volume per day, volume per week, binge drinking, and alcohol-related problems. In the coming sections, the findings of these studies will be discussed and explained thoroughly.

### 4.1. Proximity to Alcohol Outlets

One of the main factors influencing alcohol consumption is the proximity to alcohol outlets. The more outlets are made available, the easier the accessibility is to alcohol. For instance, a study by Toornstra et al. [36] reported that easy availability, low pricing, and peer pressure contribute to more alcohol consumption among adults, young adults, and adolescents in general. Among adolescents, alcohol consumption by peer access was reported to be higher with exposure to alcohol outlets [33]. In agreement with Morrison and colleagues, Martins et al. [37], in a convenience sampling method, discovered that binge drinking among students often occurs as a result of the density of alcohol outlets near the school; however, the convenience sample was not sufficient enough to be representative [75]. Correspondingly, a cross-sectional study with a randomly drawn sample from Curitiba’s public schools conducted by Cardoza et al. [41] showed an agreement with the mentioned association between adolescents’ consumption and the alcohol outlet density, where it reported that a higher density of alcohol outlets is positively associated with more consumption, and adolescents in schools located further than 250 m away from alcohol outlets were had lower consumption of alcohol. 

A similar relationship between women and alcohol outlets with alcohol consumption was also reported. Lamb et al. [35], in a cross-sectional study sampling 995 women, reported that increasing the number of alcohol outlets within as little as a 3-kilometre radius can be linked to higher levels of the harmful consumption of alcohol among women. Concordantly, Seid et al. [34] stated that more reports regarding harmful effects, such as in marriage, relationships, or finance, have been observed in women who live nearer to alcohol outlets. 

Similar associations between a higher density of alcohol outlets/liquor licenses and more alcohol consumption can be found in different countries. For instance, the density of alcohol outlets moderated the heritability of alcohol problems in a study sampling from Germany, the United Kingdom, France, and Ireland [23]; the alcohol consumption and mean daily intake of alcohol increased with the liquor license increments in two study samples from Western Australia [38,39]; the number of drinks and high consumption in general were positively associated with a high outlet density in Philadelphia, Pennsylvania [40]; and binge drinking was reported to be increasing on a daily, weekly, and yearly basis with the higher density of alcohol outlets in Mexico [76].

One recent study conducted by Mair et al. [42] in Alameda County, California, found a neutral association between the density of alcohol outlets and increased heavy drinking. However, this finding was limited to off-premise outlets and heavy drinking only. Additionally, this study lacked information about response rate, which is an important indicator of the representativity of the outcome.

Looking at the evidence, a higher density of alcohol outlets may be a contributing factor to the more frequent consumption of alcohol, which can be explained by easier sourcing of the substance. Although the outcomes and outlet types that were examined by the studies were various, this factor holds a correlation with alcohol consumption globally; however, that relation may differ between regions. For instance, in Europe, the association with increased outlets moderates the heritability of alcohol outlets; in Australia, it was related to the mean daily intake; and in North America, it was found to be related to high consumption. The unique effect of off-premise and on-premise outlets was not revealed clearly, which may require an additional investigation specified to each type of outlet. 

### 4.2. Age Exposure

Another factor related to alcohol consumption is age exposure. After being allowed to consume alcohol for the first time, adolescents are more likely to progress from drinking one drink to five or more at once [44]. Not only does early access predict consumption profiles but so does the trajectory; a study by Plenty et al. [46] showed that it is twice as likely to get high AUDIT scores and social harm and triple the odds of heavy episodic drinking for a steep escalator trajectory of young age compared to those with a slow increment of alcohol use. 

Soundararajan et al. [45] reported an association between early access to alcohol and the frequency of alcohol consumption in a study sampling 99 participants from addiction wards in India, using questionnaire data; more frequent consumption and a higher frequency of heavy consumption were positively associated. This association was also observed among students from two different nations in a study sampling 1833 13-years-old students from Washington, USA and Victoria, Australia, conducted by Kim et al. [43], who showed that the early use of alcohol predicts frequent drinking and alcohol problems later in life. 

The age at which the first alcoholic drink was consumed can be important to account for, since it may have effects on the awareness and knowledge development regarding responsible alcohol consumption. Aguilar et al. [50] observed that higher rates of drinking were among those with early onset regardless of sex. Another study that accounted for possible confounders and included parent–child dyads found that early age of onset can be linked with both frequent and infrequent binging, and initiation as early as 13 years-old or less is even more at risk of increasing frequent binging and the total volume of drinks, while later initiation (16 to 17 years old) seemed to be associated with a reduced risk of infrequent binging [52]. Additionally, Islam [51] analyzed the survey data from multiple years and found an association between age of onset and awareness of low-risk drinking and drinks counting, where a higher awareness was observed among those initiated later, especially during adolescence. These findings can be helpful, especially for parenting, since parenting authority itself has been found to have an effect, as will be discussed in the next subsection. 

As discussed above, alcohol consumption in heavy quantities might be associated with younger ages. Older ages may have less risky consumption or less consumption at all, depending on the norms and culture in the country of residence. For instance, a study conducted in Malaysia discovered that alcohol consumption odds can decrease with aging up to 0.016 times with every additional year of age [49], while, in other countries, it may be different. For instance, a study sampling from Australia, England, Scotland, New Zealand, St Kitts and Nevis, Thailand, South Africa, Mongolia, and Vietnam discovered that the drinking frequency in general increases with the age increment; however, the consumption quantity is not likely to be large, and this association was more consistent among higher-income countries [48]. In another study sampling citizens aged more than 50 years from 12 different European countries, both males and females showed a substantial relationship between age and alcohol consumption, with consumption dropping as they got older [47].

The revealed association with age may draw a timeline for the relationship between the age and consumption, where the early years of life are crucial to build an appropriate awareness about substance use, and the consumption patterns during those years may have an effect on the later stages; a younger age of exposure was related to riskier patterns and consequences, and older ages, even though they were observed to have higher consumption frequencies, showed better awareness, since the quantities were not observed to be high. 

The association with older age may differ between regions. Although the study conducted in Malaysia by Kang Cheah & Rasiah [49] revealed that the consumption odds reduced with aging, the study reported by Chaiyasong et al. [48] was done on several countries, including countries from the same region as Malaysia, which revealed the opposite association, where the frequency increased with aging. It is worth noting that the study done by Kang Cheah & Rasiah [49] used the past 30 days of consumption and consumption during the data collection as a measurement of the outcome and included all types of consumption, while the latter study used frequency, the typical occasion quantity, and volume to determine the outcome. All things considered, aging may be associated with more frequency of consumption; however, that increased frequency does not seem to be alarming, since the quantities are not likely to be large. Nevertheless, frequency and quantity both play a sensitive role at young ages, and efforts should be made for preventive measures for this population. 

### 4.3. Family Background Influence

Indeed, family has an influence in the formation of one’s personality and lifestyle and that includes the stability of the family, parenting authority, and the lifestyle of the family. In our review, we collected studies relating these factors to alcohol consumption.

At a young age, the parenting authority plays an important role, and as discussed earlier, the age of the first drink and the early access of alcohol play important roles in developing a proper awareness and drinking patterns. According to the literature, permissive parenting authority can act differently. For example, Mathialagan et al. [53] used AUDIT scores and surveys to collect data from 150 college students; the findings indicated that increased authority did not impact the consumption patterns significantly, while permissive authority was found to decrease the consumption patterns. Another study conducted by Dickens et al. [54] used survey data from 23,163 rural adolescents and found that increased parental permissiveness increased the likelihood of alcohol use the previous month, which was opposite to the earlier study. However, it is worth mentioning that the earlier study surveyed college students from SEGi College, Malaysia, which is a private university with diverse nationalities and different backgrounds, while the latest one collected survey data from middle and high school students from non-metropolitan counties in the U.S. Both studies did not include sample size justifications, and they did not measure and account for confounders such as early access or age of onset. However, the latter study used data from a larger data collection effort, which may indicate blinded assessors, which gives more strength to the latter study. Additionally, the population of the studies were different, college students from a private university vs. rural adolescents, so more studies are needed to clarify this association, and accounting for the population should take place. 

The other factor in family background is family instability. According to the literature, being widowed or divorced is linked to more problematic alcohol use [55]. Nonintact and complex family structures may be linked to alcohol misuse among adolescents [56].

Families’ alcohol dependence can also contribute. For instance, it was reported that the amount of alcohol consumed by parents and siblings has a significant impact on the amount of alcohol consumed by other siblings [59]. Additionally, having a family history of alcohol use for those with high Barratt Impulsiveness Scale scores (a questionnaire designed to assess the personality/behavioral construct of impulsiveness) was associated with severe alcohol-related consequences [57]. Living with parents with an alcohol use disorder may results in the earlier consumption of alcohol, and it was reported that male patients who lived with both parents with an alcohol use disorder were younger than the female patients and patients with parents without an alcohol use disorder when they first consumed alcohol [58]; however, this study had significantly more male participants than females, which could alternate the results when comparing males and females.

Mothers might have more impact than fathers, as one study reported that drinking increased in women who said their mother was a heavy or problem drinker and who thought they were like their mother, while the fathers’ results did not follow the same pattern [60]. In another study done by Tschorn et al. [23], early hazardous drinking (at the ages of 14–16) was linked to the mother’s exposure to alcohol during pregnancy.

In a study utilizing self-answered questionnaire sampling from grade 7 classes from private schools in Australia, neither of the above associations was discovered, only proving that accessing alcohol through a parental supply only allowed for consumption during the adolescent years without any prediction of future prevention or protection from frequent consumption, as an increase in alcohol consumption kept taking place throughout the adolesce years, regardless of the parental supply [61]. Although this study had a considerable high quality of evidence (78.57%), its findings may not be generalized for later ages, since the follow-up continued until grade 10, and nothing was reported for the later years.

The studies concerning parental authority did not yield a consistent association. Moreover, the study conducted by Clare et al. [61], despite its limited investigation timeframe, showed that an increase in consumption kept taking place regardless of the parental supply (which may indicate permissive parenting). However, the age at which parental authority is present the most showed an importance in the formation of the awareness regarding responsible alcohol consumption, as discussed in the previous subsection. Other familial factors may present an additional contribution. For instance, the consumption by parents, siblings, and historical familial exposure did show an association, with agreements across all identified studies. 

All things considered; parental permissiveness combined with the familial background may have a unique impact, since parental authority alone would not be enough to raise the awareness of children when they may imitate a familial model that was present throughout their lives. 

### 4.4. Socioeconomic Exposure

Socioeconomic exposure can be extended to the level of education, employment, type of profession, and monthly income, as all these factors may expose individuals to certain alcohol consumption patterns, including risky consumption. In terms of alcohol consumption in general, it was reported that men with a higher socioeconomic status were more likely to drink alcohol and smoke [62], professionals and mangers had more drinking occasions than semi-skilled and unskilled manual workers [29], and higher chances of alcohol consumption could be found among students who worked [41]. 

A study conducted on Malaysian participants confirmed this association, where it was found that there were 0.004 times more odds of alcohol consumption with every MYR 100 increment to the monthly income [49]. In a later study, Mair et al. [42] demonstrated the same association between high income and the frequency of consumption, as well as another finding that residents of a high-income area consume more than residents of a low-income area, regardless of the income level. Additionally, a study sampling from multiple countries discovered a similar overall association between income level and frequency of consumption, whereas high-income countries showed more frequency of consumption than middle-income countries, with the exception of one middle-income country, which is South Africa [48]. 

Although the above-mentioned studies showed a positive association between high income and consumption, other studies reported a positive association with low income and alcohol consumption. For instance, Čihák [63] studied data from statistics departments in the Czech Republic and found that economic downturns resulted in higher alcohol consumption, which could be related to the increase in the unemployment rate. Additionally, Khan & Shaw [64] discovered a higher alcohol consumption among the ST class in India (which is socially excluded and in a lower wealth category). 

The average rank of the quality of evidence for both associations was similar: 65.7% for the positive association and 67.8% for the negative; however, the number of studies was not the same, with triple the number of studies concluded with a positive association with a high income. Additionally, the Čihák [63] and Khan & Shaw [64] studies both could not be generalized, since the first one used data about alcohol-related liver cirrhosis to indicate the overall increased consumption of alcohol, which risked insufficient representation of the population, and the latter had a confounder, which was the social exclusion of the specific group, which may have shifted the findings of the study towards the group.

In general, an increased income allows individuals to afford more products, including alcoholic products, than those with limited incomes. Additionally, the time availability may increase among those with a higher socioeconomic status, especially when the professions are different. For example, professionals and managers may need to work less hours than unskilled workers, who might require more working hours to generate an income. However, students who worked were observed to be exposed to alcohol consumption; this may not be linked to more time availability but to a higher ability to afford alcoholic products. Despite that, affording the product may not be the only contributing factor, as being socially excluded or going through an economic downturn at a large scale could result in exposure, regardless of the individual’s wealth level. Nevertheless, socioeconomics can be associated with an alcohol consumption pattern; however, the degree to which this association could be present may vary according to the overall current social and economic situations of a community. 

### 4.5. Religious Influence

Religion influences individuals’ perceptions and attitudes towards alcohol and, consequently, their approach to alcohol use. Additionally, the strength of religious involvement is a big factor in protection against alcohol consumption [73]. Notably, religiosity has a negative relationship with excessive alcohol intake, whereas undergraduates who reported alcohol-related problems were found to be less religious [69]. Higher religiosity results in less odds of engaging in binge drinking [70]. Higher religiosity is also connected with less alcohol consumption frequencies for both genders [68]. Similarly, another study indicated that religiosity is linked to a reduction in alcohol usage in general, and nonbelievers consume alcohol and misuse it the most, relative to Catholics and Muslims [65]. Religious activities have been investigated as well. For instance, religious chanting/singing and reading the sacred texts are associated with lower alcohol consumption in general [72]; however, it was alco reported that frequent prayer can aid alcohol consumption reduction for moderate drinkers but not heavy drinkers [66].

One’s feelings toward religion and its importance in life may moderate the effects of religiosity. For instance, those who believed their religious heritage is prescriptive drank less alcohol and had stronger religious characteristics [72]. On the other hand, students who reported that religion played a minor role in their lives were more likely to have recently consumed alcohol [67]. Nevertheless, the presence of a religious affiliation is linked to less frequent alcohol usage [73], and religious commitment is associated with a reduced likelihood of substance misuse [74].

Religiosity may impact communities differently. For instance, in a study analyzing survey data from adolescents from rural communities, it was reported that religiosity was associated with less recent alcohol consumption but with a bigger influence on White teenagers than African Americans [54], while another study utilized data from a nationally representative sample of individuals aged 17–31 years from the U.S. and reported the same link between increased religiosity and a lower risk of any substance use; however, this link may weaken with age [74]. Basically, the effects of religious involvement can vary according to factors such as whether its involvement was during childhood or adulthood, and that too can vary according to other factors, such as ethnicity, according to Agrawal et al. [71], who reported that adulthood involvement with religion showed decreased alcohol involvement for both Black and White ethnic groups, unlike childhood involvement, which demonstrated an association only for the White group.

In general, religious involvement may be linked to more responsible alcohol consumption patterns. Although it was observed that the strength of this association may differ between populations, the exposure itself may moderate this outcome. Since more involvement with religion will yield more religious activities, such as chanting/signing and that was linked to lower alcohol consumption, this can be explained as, the larger the role religion plays in one’s life, the more effect that religion will potentially have on consumption patterns. 

## 5. Conclusions

In this paper, based on a systematic review of the extant literature, we conclude that alcohol consumption can be attributed to a range of demographic and social factors, namely access to alcohol outlets, age exposure, familial background, socioeconomic background, and religious influence. Easier access to the substance can be associated with the more frequent consumption of alcohol among adults, young adults, and adolescents in general, and it may be associated with higher levels of harmful consumption, as well as more harmful effects in life events for women. Adolescents exposed to alcohol at a young age, regardless of gender, drank more, but their drinking habits changed as they grew older. Family background influences alcohol dependence, including family stability, parental authority over early access to alcohol, and family lifestyle, which influences alcohol dependence, notably mother drinking tolerance and habit of consuming alcohol throughout pregnancy. The level of education, employment, type of profession, and monthly income can have an impact on alcohol consumption. There is a link between high religiosity and fewer alcohol-related problems, less binge drinking, and less alcohol consumption frequency for any gender. Identifying the factors associated with alcohol misuse is crucial so that the right policy and community-level prevention interventions can be provided to populations with potential substance use disorders and those affected by their negative behaviors. Understanding these factors can lead to better guidelines for alcohol use and can aid in better designs for prevention interventions.

## 6. Limitations and Outlook

The scope of this review was limited to the factors identified through the search terms, leaving behind other factors that might have had evidence of the same caliber or even greater quality. Future reviews might concentrate on addressing additional possible factors such as the effects of peer pressure and marital status. Due to the data collection being limited to the last six years (2017–2022), some significant findings may have been missed. 

Additionally, the studies identified from the search were observational studies, which can be understood, as conducting a controlled trial to measure the effect of exposure would be a huge challenge and risk exposing the research subjects to alcohol misuse.

One other limitation of our review was the influence of proximity to alcohol outlets, where we reviewed the influence of alcohol outlet density in general. However, we did not investigate the difference between on-premise and off-premise outlets, which may need further understanding. The findings in this review suggest that a wide range of factors can contribute to alcohol consumption profiles, awareness, and behaviors. 

## Figures and Tables

**Figure 1 ijerph-19-08103-f001:**
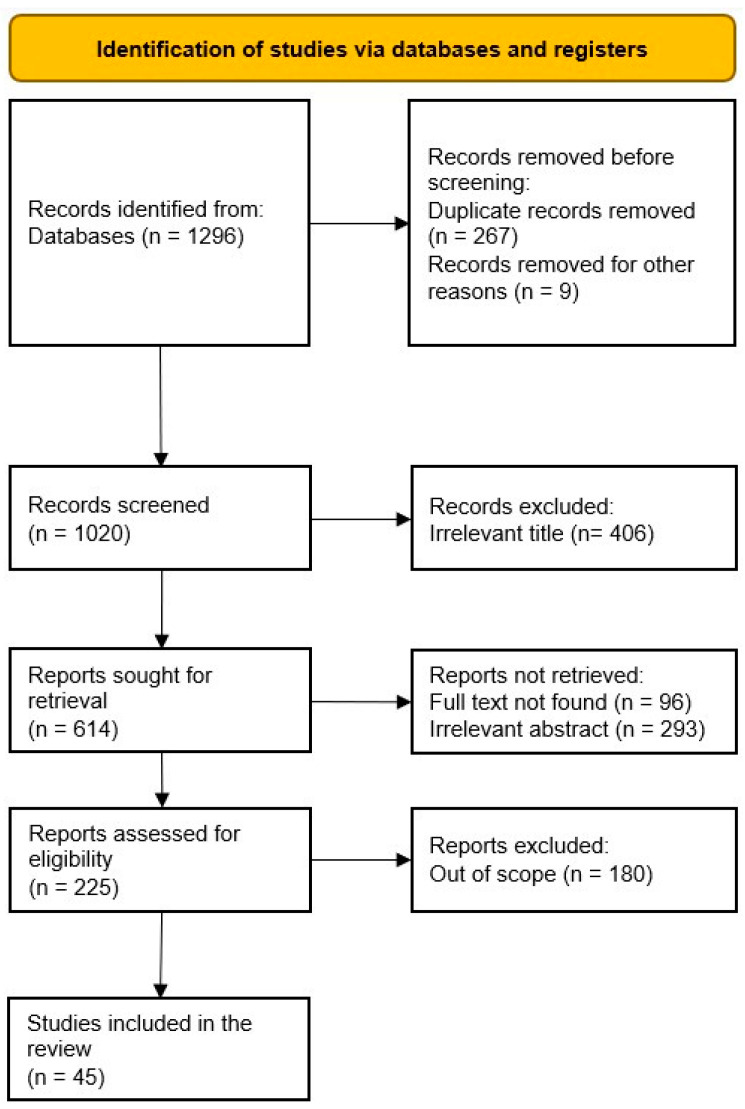
PRISMA flow diagram.

**Figure 2 ijerph-19-08103-f002:**
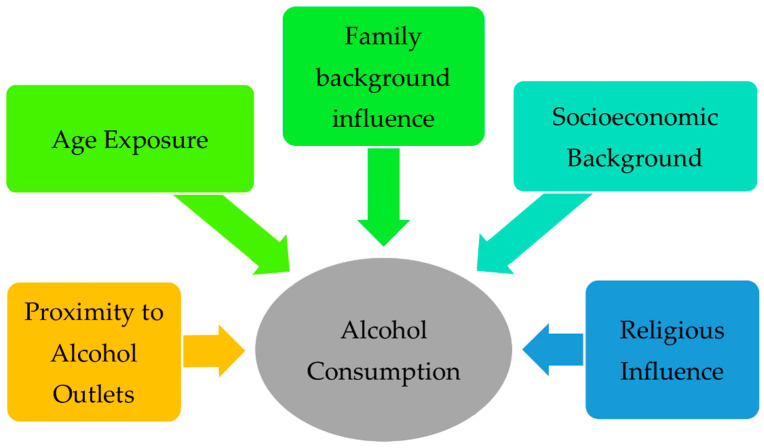
Factors associated with alcohol consumption patterns.

## Data Availability

Not applicable.

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
