# Peer review of "Alcohol Consumption Patterns: A Systematic Review of Demographic and Sociocultural Influencing Factors"

_ijerph, 2022, doi:10.3390/ijerph19138103_

Round 1
Reviewer 1 Report
This review is of great interest to public health. However, there are two issues that must be discussed by the authors to assess its possible publication:
1. Why haven't the authors used the PRISMA statement (Preferred Reporting Items for Systematic reviews and Meta-Analyses) ?
2. A good part of the factors identified are of a sociocultural nature. Authors should discuss the validity of a global analysis, versus a more regional analysis, or even by country.
Author Response
The authors would like to thank the reviewer for the comments.
Authors' responses and actions are attached.

Reviewer 2 Report
Please see attached file.

Author Response
The authors would like to thank the reviewer for the comments.
The authors' responses and actions are attached.

Round 2
Reviewer 1 Report
-
Author Response
The authors would like to thank the reviewer for reviewing this work.
Reviewer 2 Report
I appreciate the Authors' responsivity to feedback, particularly in use of person-first language and removal of causality language throughout the manuscript.
When reading the limitations section, I did not understand the following:
"Additionally, the literatures included in the qualitative synthesis were observational studies which can be understood as controlling the environment around a community over a long period; this would have been a huge challenge and risked exposing the research subjects to alcohol misuse."
I also think acknowledging the limited scope of the review is necessary in the limitations, as other factors may be equally or more important, depending on the culture and developmental period (e.g., peer use, marriage).
Thank you for allowing me to review this work.
Author Response
The authors would like to thank the reviewer for the comments. The actions are attached

This manuscript is a resubmission of an earlier submission. The following is a list of the peer review reports and author responses from that submission.